# scMPT: Towards Applying Large Language Models to Complement Single-Cell Foundation Models

## Abstract

Single-cell foundation models such as scGPT represent a significant advancement in single-cell omics, with an ability to achieve state-of-the-art performance on various downstream biological tasks. However, these models are inherently limited in that a vast amount of information in biology exists as text, which they are unable to leverage. There have therefore been several recent works that propose the use of LLMs as an alternative to single-cell foundation models, achieving competitive results. However, there is little understanding of what factors drive this performance, along with a strong focus on using LLMs as an alternative, rather than complementary approach to single-cell foundation models. In this study, we therefore investigate what biological insights contribute toward the performance of LLMs when applied to single-cell data, and introduce scMPT; a model which leverages synergies between scGPT, and single-cell representations from LLMs that capture these insights. scMPT demonstrates stronger, more consistent performance than either of its component models, which frequently have large performance gaps between each other across datasets.

## 1 Introduction

Single-cell foundation models, such as scGPT (Cui et al., 2024), have seen a surge in recent interest due to their ability to be adapted to achieve state-of-the-art performance on a variety of biological tasks. However, these models have inherent limitations. A vast amount of knowledge in the field of biology is represented as text, but these single-cell foundation models are trained only on gene expression data, and have no way to use this information. There has therefore been significant interest in applying large language models (LLMs) to single-cell transcriptomics, as many of these models have a large amount of pretrained knowledge which encompasses this knowledge of biology. LLMs could potentially leverage this knowledge to drive improvements in important tasks in single-cell analysis. Either as an alternative approach, circumventing the need to curate massive amounts of data to train new single-cell foundation models, or as a complementary approach, merging the knowledge and capabilities of LLMs and single-cell foundation models to improve performance over either.

A popular approach for enabling LLMs to work with single-cell data is converting this data to simple text sequences (i.e. cell sentences), which are encoded to generate representations that encapsulate the pre-trained knowledge of these models. This method has yielded promising results that are competitive with dedicated foundation models on certain single-cell analysis tasks such as cell type classification (Chen & Zou, 2023) (Choi et al., 2024) (Levine et al., 2023) (Fang et al., 2024). However, what pre-trained knowledge is being captured, as well as how to leverage synergies between this knowledge, and representations generated by single-cell foundation models remains largely unexplored. These questions are necessary to understand how LLMs can meaningfully improve single-cell analysis and address the shortcomings of single-cell foundation models.

In this work, we therefore use interpretability methods along with an ablation study to elucidate what pre-trained knowledge of biology LLMs leverage when applied to single-cell analysis. We then explore how these models can be used to complement single-cell foundation models in a manner that leverages synergies between representations to improve performance. Our key contributions are:

1. Despite scGPT being a state of the art foundation model, we find that fusion with LLMs can improve upon its performance, indicating that cell representations derived from text and single-cell data are complementary. We introduce scMPT, which leverages synergies between representations generated by scGPT and an Ember-V1 text encoder, enabling better overall performance.

2. We find LLMs interpret and represent cell sentences in a way that leverages biological insight, and specifically knowledge of marker genes.

## 2 RELATED WORK

### 2.1 LARGE LANGUAGE MODELS

LLMs have received much attention due to their versatility and strong performance across a variety of domains and tasks. They have been shown to perform well on classification, question-answering, fact retrieval, and more, even without fine-tuning (Gallegos et al., 2024). These models, typically based on the Transformer architecture (Vaswani, 2017), consist of millions or even billions of trainable parameters, and are pre-trained with vast amounts of language data which often encompasses many domains, facilitating this versatility (Zhang et al., 2024a).

LLMs are often used to generate text in an autoregressive fashion, or to generate representations of text in the form of embeddings that can be used for a variety of downstream tasks. However, models used for each of these tasks are generally quite different, with LLMs designed for text generation typically employing a different type of architecture than text embedding models (Zhang et al., 2024a). We will therefore study each type of model separately.

### 2.2 SINGLE-CELL FOUNDATION MODELS

Inspired by the success of LLMs, single-cell foundation models such as scGPT have been developed that display broad capabilities across many biological tasks such as cell type annotation and multi-batch integration (Chen & Zou, 2023) (Cui et al., 2024). scGPT is, like most LLMs, based on the transformer architecture. Key to its development was curating and pre-training on a massive amount of single-cell data, specifically from over 33 million cells from CELLxGENE (Cui et al., 2024) (Megill et al., 2021).

### 2.3 APPLYING LARGE LANGUAGE MODELS TO SINGLE-CELL ANALYSIS

To enable large language models to work with single-cell data, existing studies generally represent this data as text. Perhaps the most common representation used, which we will focus on in this study, is the "cell sentence"; a textual sequence which lists gene names in descending order of expression level for a given cell. For example, *"A cell with genes ranked by expression: RAB3B MT-CO1 CHN1 HNRNPA1P40 SYT1....."*. The Cell2Sentence paper demonstrated that conversion to this representation incurs minimal information loss. This was accomplished through training a linear model to accurately predict gene expression from gene rank, and motivates our focus on this method (Levine et al., 2023). Studies that use LLMs with cell sentences for single-cell analysis broadly fit into two categories; those that use generative models, and those that use embedding models.

Studies that explore the use of generative models include Cell2Sentence (Levine et al., 2023), and "How do Large Language Models understand Genes and Cells" by Fang et al. (2024). The results presented in the latter fell short of scGPT on tasks such as cell type annotation, despite using cell sentences to fine tune large LLMs with up to 13 billion parameters; a process which is quite computationally expensive. While Cell2Sentence performance is much more competitive with single-cell foundation models such as scGPT, this method requires curating large single-cell datasets for a fine-tuning process which involves multiple stages and is computationally expensive. However, ideally this would not be necessary when working with LLMs due to the knowledge of biology they obtain during pre-training, enabling "off-the-shelf" usage. Although not the primary focus of this work, the scELMO paper by Liu et al. (2023) evaluates the zero-shot cell type annotation performance of GPT-4 when using cell sentences, and reports that the method is completely ineffective, yielding an accuracy of 0 % on all datasets tested.

Studies that use embedding models include GenePT (specifically, the GenePT-s approach) (Chen & Zou, 2023), and CELLama (Choi et al., 2024). Both papers reported performance that was competitive with scGPT on tasks such as cell type classification in a zero-shot setting. However, a limitation of these works is that they provide little justification for their selection of embedding models, despite these models often varying widely in performance on different tasks (Muennighoff et al., 2022). The focus of these works is also primarily on how LLMs can be used as an alternative to single-cell foundation models, rather than in a complementary fashion. The GenePT paper does notably experiment with an ensemble approach that aggregates the nearest neighbours from GenePT-s, scGPT, as well as their GenePT-w method to make a final cell type prediction (Chen & Zou, 2023). However, fusion at such a late stage ignores possible synergies between different modalities (Steyaert et al., 2023).

## 3 METHODS

### 3.1 DATA COLLECTION AND TRANSFORMATION

We focus our experiments on the datasets that were used to evaluate the cell type classification and clustering performance of GenePT (Chen & Zou, 2023), as well as the subsample of the Tabula Sapiens dataset (Consortium* et al., 2022) used to evaluate CELLama (Choi et al., 2024). For each dataset, we use the same train/test split as each of these respective works to evaluate performance on downstream tasks. Cells were represented using cell sentences following the same approach as GenePT-s (Chen & Zou, 2023), where gene names are listed in descending order of expression level, omitting genes with zero counts. These cell sentences are then passed to text encoders to generate cell embeddings, or to generative LLMs for cell type classification through autoregressive generation.

### 3.2 CELL EMBEDDING APPROACHES

In general, text encoders can vary greatly in performance on tasks such as classification and clustering. To determine what LLM of this type to use for our experiments, we therefore test a variety of pre-trained models selected using MTEB (Muennighoff et al., 2022). We use the same experimental design and metrics as the GenePT paper to evaluate classification and clustering performance (Chen & Zou, 2023). For example, using a k-nearest neighbours method for zero-shot cell type classification. Details can be found in Appendix A.1.

As another experiment, we train a small multi layer perceptron (MLP) on top of text encoders used to generate cell embeddings as an alternative to the 10-nearest neighbour classifier for cell type classification. This setup has the potential to improve performance with minimal training, but more importantly, the differentiability of the MLP facilitates a wider range of interpretability methods than k-nearest neighbours. We simply use the default architecture for the scikit-learn library's MLP implementation (Pedregosa et al., 2011), and leave the text encoder frozen during training to reduce computational cost. We report accuracy along with macro-weighted precision, recall, and F1 score for all datasets.

Building on this idea of training an MLP on top of the frozen text encoder, we train a multimodal network for cell type classification on top of our top performing text encoder and scGPT, which we coin scMPT. scMPT combines extracted features from each encoder, and aims to leverage potential synergies between representations. Both encoders are notably left frozen during training to reduce computational cost, and maintain the domain specific knowledge encoded in each model. We report accuracy along with macro-weighted precision, recall, and F1 score for all datasets. The simple architecture of scMPT takes inspiration from works such as Kwak et al. (2023) and Miller et al. (2020), and is presented in Figure 1 below.

### 3.3 GENERATIVE APPROACHES

To classify individual cells using generative LLMs, we pass in the cell sentence for the cell we want to classify, and instruct the LLM to output the most likely cell type given a list of all cell types from the corresponding train set. Providing this list of cell types, a notable change from previous work, is necessary to have a fair comparison of classification performance with scGPT or text embedding methods, since k-nearest neighbours will also only output labels present in the train

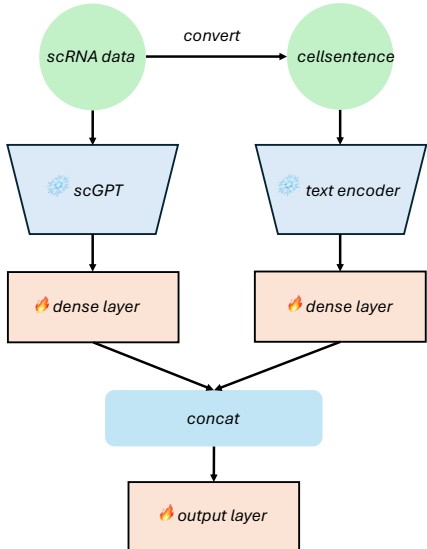

Figure 1: A depiction of the scMPT architecture. Cell embeddings from scGPT and a text encoder are fed into dense layers. The output of these dense layers is then concatenated, and fed into an output layer which predicts final cell type. Encoders are left frozen, while dense and output layers are trainable.

set. To limit costs, we focus on the three datasets used to evaluate cell-type annotation in the scGPT paper, namely the Myeloid (Cheng et al., 2021), Pancreas (Luecken et al., 2022), and Multiple Sclerosis (MS) (Schirmer et al., 2019) datasets. For each dataset we evaluate cell type classification performance on a subset of 100 randomly selected cells from the test set, reporting accuracy.

As an alternative multimodal approach to scMPT, we then investigate whether GPT-4o can be used to complement scGPT, leveraging the domain specific knowledge encoded in each model to improve classification performance. Our pipeline is constructed as follows: For a given cell we want to classify, we determine the three most likely cell types using scGPT and the 10-nearest neighbour method of cell type classification previously described. We then employ a two API call setup to determine which of these three cell types is most likely correct using GPT-4o. Specifically, in our first API call, we pass in the cell sentence for the cell we want to classify, along with the list of three potential cell types, and instruct the LLM to generate reasoning about the most likely cell type. We then feed this reasoning to GPT-4o in our second API call to generate the final cell type. We adopt a two-stage setup based on previous results that suggest it can improve LLM reasoning when working with multimodal information (Toma et al., 2024) (Zhang et al., 2023). We instruct GPT-4o to pick the first cell in the list, which is the most likely class according to scGPT, if it is uncertain. Prompts used can be found in Appendix A.3.

### 3.4 INVESTIGATING WHAT FACTORS CONTRIBUTE TOWARD LLMS' PERFORMANCE

To investigate the factors contributing to the competitive performance of LLMs, specifically focusing on text encoders, we first investigate what features from cell sentences are focused on by the model when predicting cell type. We employ two interpretability techniques. The first, integrated gradients, is a gradient based method which has seen significant recent adoption in the biomedical domain, including for interpreting language models (Sundararajan et al., 2017) (Talebi et al., 2024). The second, Local Interpretable Model Agnostic Explanations (LIME) is a model agnostic interpretability method that has also seen significant usage in this domain (Ribeiro et al., 2016) (Wu et al., 2023) (Laatifi et al., 2023). We apply both interpretability techniques on the model that consists of an MLP trained on top of our highest performing frozen text encoder. We focus on this model rather than the setup that uses k-nearest neighbours since the MLP is differentiable, facilitating the usage of integrated gradients. We limit our analysis to cell types that are specific and have clearly defined names. For each dataset and each interpretability method, we calculate feature

attributions for ten cells of each type (or all cells if there are less than ten in the test set). We then sum attributions across the ten cells of each type, and examine the top ten genes with the highest positive attribution scores to determine what genes influence the model to predict a cell is of a given type.

We also conduct a series of ablation tests to determine whether there are factors contributing to the performance of text encoders that are not related to their knowledge of biology. We accomplish this through ablating each major element of biological information in the cell sentences that was derived from the raw single-cell data, namely the gene names, and the ordering of gene names which was based on expression level. We then observe how this impacts clustering and classification performance. Specifically, for our first ablation test, we replace gene names in cell sentences with unique hashes generated using the SHA-256 algorithm, truncated to ten characters. We then investigate the effect of shuffling the order of gene names within cell sentences. Finally, we investigate the effect of applying both of these ablations.

# 4 RESULTS

Previous work on using LLMs with cell sentences has not considered the large variety of text encoders available which vary substantially in performance, and has not been able to successfully use generative LLMs with cell sentences in a zero-shot setting. We therefore address each of these major gaps before proceeding with our main experiments. We then investigate what biological insights, and other factors, contribute to the performance of LLMs in single-cell analysis through the use of interpretability and ablation tests. Finally, we introduce scMPT, which leverages synergies between embeddings generated by LLMs and scGPT. We also introduce an alternative fusion method which uses GPT-4o to complement scGPT.

## 4.1 ENCODERS USED IN PREVIOUS WORK ARE OUTPERFORMED BY EMBER-V1

To select an encoder for our main experiments, we investigate the performance of different pre-trained LLMs for generating cell embeddings from cell sentences.

We first compare the cell type classification and clustering performance of all encoders of potential interest on the Aorta dataset (Li et al., 2020). Specifically, we compare the performance of six encoders selected for their performance on the MTEB benchmark against all-MiniLM-L12-v2 (hug, b) and OpenAI ada-002 (OpenAI), text encoders which previous work has focused on for generating cell embeddings. We also compare against OpenAI text-embedding-3-large (OpenAI), and scGPT (Cui et al., 2024). Results are presented in Table 1. We observe that ada-002 and all-MiniLM-L12-v2 were both outperformed by several of the encoders selected from MTEB on both cell type classification and clustering. Ember-V1 performed particularly well, outperforming both of these encoders by a wide margin and closing much of the gap between text encoder performance and scGPT performance on this dataset. We also observe that text-embedding-3-large performed significantly worse than ada-002, despite it being a newer text embedding model from OpenAI designed to improve performance (OpenAI).

We then compare Ember-V1 against ada-002 and allMiniLM-L12-V2 on all other datasets collected. This includes the MS (Schirmer et al., 2019), Artery (Alsaigh et al., 2022), Bones (Chou et al., 2020), Myeloid (Cheng et al., 2021), Pancreas(Luecken et al., 2022), and subsampled Tabula Sapiens (Consortium* et al., 2022) datasets. Results for classification and clustering performance are reported in Tables 9,10,11,12,13, and 14 (see Appendix A.1). Overall, Ember-V1 outperforms allMiniLM-L12-V2 on every metric on every dataset. The improvement in performance over ada-002 is more modest, with improved clustering and classification performance on 5/7 datasets tested overall. However, ada-002 is not an open source model, which makes many interpretability methods challenging or impossible to use.

Ultimately, we find that switching pre-trained text encoders to Ember-V1 can lead to improved performance over previous methods to generate cell embeddings from cell sentences, motivating us to select this encoder for our main experiments.

Table 1: Zero-shot cell type classification and clustering performance of encoders on the Aorta dataset. Highest value for a metric in bold, second highest underlined.

| Model | Accuracy | Precision | Recall | F1 | k-means ARI | k-means AMI |
|---|---|---|---|---|---|---|
| ada-002 | 0.872 | 0.865 | 0.670 | 0.716 | 0.350 | 0.510 |
| text-embedding-3-large | 0.791 | 0.570 | 0.438 | 0.451 | 0.160 | 0.200 |
| all-MiniLM-L12-v2 | 0.855 | 0.731 | 0.623 | 0.644 | 0.441 | 0.495 |
| Ember-V1 | 0.906 | 0.910 | 0.800 | 0.841 | **0.535** | 0.597 |
| gte-large-en-v1.5 | 0.894 | 0.903 | 0.746 | 0.791 | 0.442 | 0.529 |
| mxbai-embed-large-v1 | 0.901 | 0.885 | 0.761 | 0.801 | 0.303 | 0.506 |
| bge-large-en-v1.5 | 0.905 | 0.902 | 0.799 | 0.837 | 0.405 | 0.546 |
| GIST-small-embedding-v0 | 0.871 | 0.859 | 0.696 | 0.735 | 0.342 | 0.482 |
| stella_en_400m_v5 | 0.905 | 0.903 | 0.804 | 0.838 | 0.323 | 0.552 |
| scGPT | **0.960** | **0.958** | **0.942** | **0.949** | 0.463 | **0.637** |

## 4.2 Providing Possible Labels Can Dramatically Improve the Zero Shot Cell Type Classification Performance of Generative LLMs

We next investigate whether there are simple-to-implement changes that can improve the zero shot cell type classification performance of generative LLMs when working with cell sentences compared to previous results, to a point where further experiments using these methods have value.

We evaluate the cell type classification performance of GPT-4o when passing in the cell sentence for the cell of interest, and instructing the LLM to output the most likely cell type. Different from previous work, we also pass in a list of potential cell types derived from the train set, aiming to improve performance, and have a setup that can be more fairly compared with models like scGPT. We compare performance against another state-of-the-art LLM, Llama 3.1 405B (Dubey et al., 2024), and also report the performance of scGPT as a reference. We report classification accuracy for all models on a subset of 100 cells from each of the Pancreas, Myeloid, and MS datasets in Table 2.

Table 2: Cell type classification accuracy of generative LLMs on different datasets when provided with a list of possible labels. scGPT included for reference.

| Dataset | Model | Accuracy |
|---|---|---|
| Pancreas | GPT-4o | 0.96 |
| | Llama 3.1 405B | 0.80 |
| | scGPT | 0.77 |
| Myeloid | GPT-4o | 0.34 |
| | Llama 3.1 405B | 0.29 |
| | scGPT | 0.51 |
| MS | GPT-4o | 0.34 |
| | Llama 3.1 405B | 0.33 |
| | scGPT | 0.74 |

In general, the performance of GPT-4o was reasonably strong. While this model was outperformed overall by scGPT, the performance was at least comparable, with GPT-4o performing much better on the Pancreas dataset. Performance was also consistently stronger than Llama 3.1 405B, although this model still performed reasonably well. This is a dramatic improvement compared to previously reported results, which found that GPT-4 was unable to classify any cells correctly (Liu et al., 2023). We also attempt to replicate these results by repeating our evaluation of GPT-4o without passing in a list of potential cell types, and find the model to be completely ineffective, achieving an accuracy of 0 % and corroborating previous findings.

Therefore, we find that simply passing in a list of potential cell types can lead to a dramatic improvement in the zero-shot cell type classification performance of generative LLMs compared to previous results, justifying further experimentation with these models.

### 4.3 Language Models Effectively Leverage Knowledge of Marker Genes - However, Factors Unrelated To Biology May Also be Contributing Toward Their Strong Performance

To determine what factors contribute toward the strong performance of language models, and in particular text encoders, we conduct a series of interpretability and ablation tests. We focus on text encoders as this form of LLM has proven to be truly competitive with scGPT on certain single-cell analysis tasks, and models in this class tend to be small enough that a wider range of interpretability methods are computationally feasible.

The interpretability methods we use are integrated gradients, and LIME. For this analysis we focus on the Pancreas and Aorta datasets, given the strong performance of Ember-V1 on them. We apply both interpretability methods to a model consisting of an MLP trained on top of a frozen Ember-V1 encoder to determine what parts of cell sentences contribute toward the prediction of certain cell types. Specifically, through calculating attribution scores for different gene names, which show how much those gene names contributed toward predicting a cell was of a given type. Early in this analysis, we noticed that several of the genes corresponding to top attribution scores were marker genes. We therefore used PanglaoDB (Franzén et al., 2019) to investigate what marker genes were represented in the lists of top attribution scores for each cell type, calculated through aggregating results from multiple cells of a given type for a more global level of insight. Results are presented in table 3 and 4. For each cell type in the Pancreas dataset, both methods had multiple markers within the top ten genes with the highest attribution scores. There was also at least one and often multiple markers highlighted by both interpretability methods for each cell type, with this agreement more strongly indicating these marker genes were focused on by the model. For the Aorta dataset, LIME once again had multiple markers within the top ten most important genes for each cell type. While integrated gradients had less, it is notable that the markers highlighted by integrated gradients were almost an exact subset of the markers highlighted by LIME, suggesting these markers were focused on by the model. Ultimately, the representation of marker genes among the genes with the highest attribution scores for each cell type, along with the level of agreement between interpretability methods indicates that the model focuses on marker genes when predicting cell type, contributing to its performance being competitive with scGPT.

Table 3: Marker genes in top 10 gene attributions for Ember-V1 encoder +MLP with different interpretability methods - Pancreas dataset.
*(Markers that were highlighted by both interpretability methods in bold)*

| Cell Type | LIME | Integrated Gradients |
|---|---|---|
| **Alpha** | **GCG**, **TTR**, CRYBA2, TM4SF4, LOXL4 | **GCG**, **TTR**, NEUROD1, GC, ALDH1A1, SLC30A8 |
| **Beta** | INS, **IAPP**, **ADCYAP1** | **ADCYAP1**, **IAPP** |
| **Gamma (PP)** | **PPY**, ISL1 | **PPY**, NEUROD1 |
| **PSC** | **COL6A3**, FN1, TIMP1 | COL1A1, COL1A2, **COL6A3**, COL3A1 |
| **Ductal** | SERPING1, **KRT19**, MUC1 | **KRT19**, MMP7, SERPINA3 |
| **Endothelial** | **PLVAP**, **RGCC**, PODXL | **PLVAP**, **RGCC**, IGFBP7 |
| **Epsilon** | **GHRL**, **S100A6**, **SPINK1**, ACSL1 | **GHRL**, **SPINK1**, **S100A6**, HMGCS2 |
| **Mast** | **TPSAB1**, CPA3 | **TPSAB1**, ICT4S |
| **Acinar** | **PRSS1**, **CXCL17**, **PRSS3**, **REG1A** | **PRSS1**, **PRSS3**, **REG1A**, CELA3A, **CXCL17**, CELA2A |
| **Delta** | **SST**, **RBP4**, LEPR, PCSK1 | **SST**, **RBP4** |

For our ablation tests, we experiment with ablating the major elements of biological information in cell sentences; the gene names, which are replaced with truncated SHA-256 hashes, and their order, which is shuffled. We report how these ablations affect performance on the Aorta dataset for Ember-V1 in table 5 below, and ada-002 in table 21 (see Appendix A.4), using the k-nearest neighbours setup for cell type classification. Note that to be able to compare the effect of ablations

Table 4: Marker genes in top 10 gene attributions for Ember-V1 encoder + MLP with different interpretability methods - Aorta dataset.
*(Markers that were highlighted by both methods in blue)*

| Cell Type | LIME | Integrated Gradients |
|---|---|---|
| NK | **KLRB1**, **NKG7**, **XCL1**, XCL2, KLRD1 | **KLRB1**, **XCL1**, **NKG7** |
| T Cell | S100A4, TNFAIP3 | |
| B Cell | **HLA-DRA**, CD74, IGKC | **HLA-DRA** |
| Fibroblast | **MGP**, CTGF, LUM, COL1A2, DCN | IGFBP6, **MGP** |
| Mast Cell | **TPSAB1**, TPSB2, RGS1, KIT | **TPSAB1** |
| Plasma Cell | **IGHG1**, **IGKC**, **IGHG3**, IGHA1, **IGHG4**, IGLC2, JCHAIN | **IGHG3**, **IGKC**, **IGHG4**, **IGHG1** |

on each encoder's performance, we truncate the cell sentences encoded by ada-002 to be the same length as Ember-V1, which has a shorter context length. We find that both encoders, and especially ada-002, had only a moderate drop in performance when ablating gene names. To confirm this is not because of unique characteristics of the Aorta dataset, we also see how this ablation affects the cell type classification performance of Ember-V1 on all other datasets, finding a consistently moderate drop in performance as can be seen in Appendix A.4. The effect of shuffling gene names on the performance of Ember-V1 was also quite moderate, but was more significant for ada-002. However, given the smaller drop in performance for ada-002 than Ember-V1 when ablating gene names, this is likely because the encoder places higher value on syntactical similarity, rather than being because of additional knowledge of biology. The combination of both ablations lead to a large drop in performance for both encoders, however, it is notable that both encoders still retained some performance. Therefore, there may be factors unrelated to biology contributing to the performance of text encoders.

Table 5: Ablation test results - Ember-V1 encoder, Aorta dataset (zero-shot)

| Metric | Ember-V1 Gene Ablated | Ember-V1 Gene+Order Ablation | Ember-V1 Order Ablation | Ember-V1 |
|---|---|---|---|---|
| **Accuracy** | 0.830 | 0.615 | 0.856 | 0.906 |
| **Precision** | 0.816 | 0.286 | 0.902 | 0.910 |
| **Recall** | 0.563 | 0.221 | 0.623 | 0.800 |
| **F1** | 0.605 | 0.216 | 0.681 | 0.841 |
| **k-means ARI** | 0.360 | 0.092 | 0.506 | 0.535 |
| **k-means AMI** | 0.390 | 0.081 | 0.529 | 0.597 |

Ultimately, our interpretability tests strongly suggest that text embedding models leverage marker gene knowledge in cell type prediction, especially given the level of agreement between LIME and integrated gradients. However, the ablations we conducted indicate that factors unrelated to biology may be contributing to the performance of text encoders as well.

### 4.4 SCMPT AND SIMPLE TWO-STAGE LLM PIPELINES CAN ENABLE LLMS TO COMPLEMENT SCGPT, IMPROVING PERFORMANCE

Finally, we investigate how LLMs can be leveraged to more effectively complement deep learning models trained on single-cell data.

We first investigate how Ember-V1 can be used with scGPT in a complementary fashion. For each dataset, we use the train split to train a simple multimodal neural network on top of these encoders, which are left frozen. We then evaluate cell type classification performance on the test split. To provide a baseline, we also evaluate the performance of training an MLP on top of each individ-

ual encoder. We report results for the Pancreas dataset in table 6 and results for other datasets in Appendix A.2. We observe that the fusion model, which we coin scMPT, performs competitively with, and often better than the best of the two encoders on each dataset. The performance of scMPT is notably strong enough that it is even competitive with full fine tunes of scGPT on each dataset, based on results reported in the original scGPT paper (Cui et al., 2024). We also observe that simply training an MLP on top of scGPT or Ember-V1 performs quite well, and can improve performance significantly over k-nearest neighbours. This is shown in Figure 2 and Figure 3 (see Appendix A.2).

Table 6: Cell type classification performance of scMPT vs. unimodal models on the Pancreas dataset. scMPT results reported as Mean (Standard Deviation)

| Model | Accuracy | Precision | Recall | F1 |
|---|---|---|---|---|
| **scMPT** | 0.962 (0.0019) | 0.764 (0.0017) | 0.752 (0.0037) | 0.745 (0.0039) |
| **scGPT (full fine-tune - reported)** | 0.968 | 0.735 | 0.725 | 0.718 |
| **Ember-V1 + MLP** | 0.974 | 0.6815 | 0.694 | 0.684 |
| **scGPT (from-scratch - reported)** | 0.936 | 0.665 | 0.668 | 0.622 |
| **scGPT + MLP** | 0.865 | 0.625 | 0.614 | 0.592 |

As an alternative to scMPT, we next investigate how GPT-4o can be used to complement scGPT. Specifically, through using the LLM to guide the predictions of scGPT, narrowing down the top three cell types predicted as most likely for a cell of interest to a final prediction based on the cell's cell sentence. We report results on a subset of 100 cells from each of the Pancreas, Myeloid, and MS datasets, and include the accuracy of scGPT and GPT-4o as a baseline, in table 7 below. We find that our method that combines the two models performs competitively with the best model for a given dataset.

Table 7: Accuracy of scGPT, GPT-4o, and fusion method.

| Model | Pancreas Data | Myeloid Data | MS Data |
|---|---|---|---|
| scGPT | 0.77 | 0.51 | 0.74 |
| GPT4o | 0.96 | 0.34 | 0.34 |
| GPT4o+scGPT | 0.93 | 0.54 | 0.72 |

Overall, we find methods that use LLMs to complement scGPT can yield cell type classification performance that is competitive with, and often better than the more performant of the two component models. scMPT, which uses Ember-V1 to complement scGPT performs particularly well. We also find that training a small MLP on top of Ember-V1 or scGPT can lead to improved cell type classification performance over k-nearest neighbours.

## 5 DISCUSSION

LLMs have shown great potential in single-cell analysis, often performing competitively with dedicated state of the art foundation models. In this study, after finding simple methods of improving this performance, we obtain an understanding of the biological insight and other factors contributing to it, and finally explore how LLMs can better be used to complement rather than compete with single-cell foundation models, culminating in our introduction of scMPT.

We find that simply switching the pre-trained encoder used to Ember-V1 can yield significant improvements over the previously proposed GenePT-s and CELLama methods, illustrating the importance of testing different encoders. For generative methods, it was interesting to see that restricting the output space of GPT-4o by providing a list of possible cell types improved performance so dramatically compared to results from prior work. This shows that large generative LLMs can be grounded effectively with cell sentences, potentially opening up new possibilities for how these LLMs like GPT-4o can be used in single-cell analysis.

Ember-V1's focus on marker genes when predicting cell type was intriguing, as focusing on markers is a common approach for both automatic and manual cell type annotation (Clarke et al., 2021). It is therefore highly encouraging that this is a factor contributing towards the model's predictions, and ultimately its performance. One important limitation of our analysis, however, is that integrated gradients and LIME are local interpretability methods. While aggregating attribution scores across many examples can help get a more global understanding of the model's behaviour, and has been used in previous work (Talebi et al., 2024), this method is still limited. We therefore take inspiration from the analysis of Liétard et al. (2021) to get a more global understanding of Ember-V1 and its knowledge of marker genes. For each dataset, we compute the cosine similarity between the top marker gene name for each cell type from PanglaoDB (Franzén et al., 2019) and other marker genes from the same cell, and compare this to the cosine similarity between these top markers and marker genes of different cells, averaging results over all cell types. As shown in table 8, we find that the cosine similarities to marker genes of the same cell type (Intra-Similarity) are higher, providing further evidence that the model is able to leverage knowledge of marker genes.

Table 8: Cosine similarity between top marker gene names and markers of same cell type (Intra-Similarity), different cell types (Inter-Similarity), and gap between these.

| Dataset | Intra-Similarity | Inter-Similarity | Gap |
|---------|------------------|------------------|-------|
| Pancreas | 0.644 | 0.623 | 0.021 |
| Aorta | 0.667 | 0.653 | 0.014 |

With that said, based on ablation studies, there may be other factors contributing to the performance of text embedding models that are unrelated to biology. As previously mentioned, ada-002 likely values syntactic similarity between cell sentences highly. This is based on the large drop in performance when shuffling gene names, but moderately small drop in performance when ablating gene names (where these ablated gene names would notably still be in the same order). Ember-V1 interestingly had a moderate drop in performance both when ablating gene names or shuffling gene order. A potential explanation for this is that cells of the same type likely have higher overlap between their top genes expressed (which are the only ones present in the truncated cell sentence) compared to cells of different types; lexical similarity which Ember-V1 may be leveraging. This is also a potential explanation for why encoders still maintained some performance even when combining both ablations. We therefore caution that high classification and clustering performance alone does not demonstrate an encoder has a strong understanding of the underlying biology of cell sentences. With that said, we also note that an encoder placing some value on syntactic or lexical similarity is not strictly a flaw in this context. The highest expressed genes and their rank for different cells of a given type may indeed be quite similar, which would lead to a high degree of lexical and syntactic similarity. If a text encoder exploited this simple structure to outperform models such as scGPT, this would not necessarily be undesirable.

The fusion methods evaluated, which were designed to more effectively leverage synergies between modalities than prior work, performed quite well, especially scMPT. On datasets where both modalities yielded similar performance individually, the performance of scMPT was in general modestly stronger than either individual modality. However, potentially the more valuable result was that on datasets where one modality yielded poor performance, scMPT still performed well. This consistency in performance is valuable because performance between the two modalities varied widely on certain datasets. For example, Ember-V1 outperformed scGPT on the Pancreas and Bones datasets, whereas scGPT outperformed Ember-V1 by a wide margin on the Multiple Sclerosis dataset.

One important direction for future research would therefore be to test the performance of text embedding models and scGPT on many more datasets to gain a more comprehensive understanding of when each type of model may perform better. Another potential direction for future research, which could benefit from this, is the development of more advanced fusion methods. For example, large multimodal models have shown strong multimodal classification performance, and would be an interesting direction to explore (Alayrac et al., 2022). Finally, a limitation of this study is that only cell sentences were focused on as textual representations of single-cell data. In future work, it may be valuable to investigate whether there are other textual representations of single-cell data that can drive stronger performance.

## 6 REPRODUCIBILITY STATEMENT

References to all datasets, and pre-trained models used are provided in this work. The architecture of any models trained is specified, with additional details for scMPT provided in Appendix A.2. Prompts used for the method that fused GPT-4o and scGPT are provided in Appendix A.3.

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

# A  Appendix

## A.1  Zero Shot Performance of Various Encoders - Additional Details and Results

We test a variety of pre-trained text encoders for generating cell embeddings from cell sentences, selected using the Massive Text Embedding Benchmark (MTEB) (Muennighoff et al., 2022) (https://huggingface.co/spaces/mteb/leaderboard). Encoders were selected through looking at classification performance on this benchmark, filtering out encoders with over one billion parameters, or that were proprietary. We then select from the top ten encoders after applying these criteria, taking only the top encoder from a given family if several encoders from this family appear, and omitting encoders that were not compatible with the sentence-transformers library. The final encoders selected from MTEB were stella-en-400M-v5 (hug, a), gte-large-en-v1.5 (Zhang et al., 2024b) (Li et al., 2023), GIST-small-Embedding-v0 (Solatorio, 2024), ember-v1 (Nur & Aliyev, 2023), bge-large-en-v1.5 (Xiao et al., 2023), and mxbai-embed-large-v1 (Li & Li, 2023) (Lee et al., 2024). These encoders were tested against all-MiniLM-L12-v2 (the encoder used for CELLama), OpenAI ada-002 (the encoder used for GenePT), and OpenAI's newest text embedding model text-embedding-3-large.

To assess the cell type classification and clustering performance of various text encoders, we base our experimental design on the GenePT paper. For classification, we apply a 10-nearest neighbour classifier, classifying each cell in the test set of a given dataset based on the labels of its 10-nearest neighbours from the corresponding train set, measured using cosine similarity between cell embeddings. Accuracy, along with macro-weighted precision, recall and F1 score are then reported. To evaluate cell type clustering, for each encoder on each dataset, we apply k-means clustering on the cell embeddings, setting the number of clusters to match the number of cell types for the given dataset. We then compute the Adjusted Rand Index (ARI) and Adjusted Mutual Information (AMI) to evaluate the concordance between the resultant clusters and the true cell type labels. Additional results for this experiment are reported below.

Table 9: Zero-Shot cell type classification and clustering performance of different encoders on the Myeloid dataset.

| Metric | ada-002 | Ember-V1 | scGPT | all-MiniLM-L12-v2 |
|---|---|---|---|---|
| Accuracy | 0.518 | 0.499 | 0.545 | 0.497 |
| Precision | 0.359 | 0.333 | 0.336 | 0.330 |
| Recall | 0.287 | 0.266 | 0.294 | 0.254 |
| F1 | 0.306 | 0.284 | 0.306 | 0.268 |
| K-means ARI | 0.297 | 0.283 | 0.414 | 0.241 |
| K-means AMI | 0.393 | 0.409 | 0.516 | 0.304 |

Table 10: Zero-shot cell type classification and clustering performance of different encoders on the MS dataset.

| Metric | scGPT | Ember | ada-002 | all-MiniLM-L12-v2 |
|---|---|---|---|---|
| Accuracy | 0.752 | 0.444 | 0.460 | 0.416 |
| Precision | 0.667 | 0.457 | 0.467 | 0.415 |
| Recall | 0.616 | 0.338 | 0.380 | 0.331 |
| F1 | 0.596 | 0.325 | 0.360 | 0.319 |
| K-means ARI | 0.292 | 0.185 | 0.247 | 0.168 |
| K-means AMI | 0.480 | 0.394 | 0.339 | 0.298 |

Table 11: Zero-shot cell type classification and clustering performance of different encoders on the Pancreas dataset.

| Metric | scGPT | Ember-V1 | ada-002 | all-MiniLM-L12-v2 |
|--------|-------|----------|---------|-------------------|
| Accuracy | 0.784 | 0.984 | 0.983 | 0.978 |
| Precision | 0.594 | 0.796 | 0.805 | 0.770 |
| Recall | 0.550 | 0.767 | 0.742 | 0.667 |
| F1 | 0.545 | 0.778 | 0.769 | 0.699 |
| K-means ARI | 0.202 | 0.864 | 0.487 | 0.456 |
| K-means AMI | 0.411 | 0.827 | 0.740 | 0.666 |

Table 12: Zero-Shot cell type classification and clustering performance of different encoders on the bones dataset.

| Metric | ada-002 | Ember-V1 | scGPT | all-MiniLM-L12-v2 |
|--------|---------|----------|-------|-------------------|
| Accuracy | 0.353 | 0.427 | 0.326 | 0.311 |
| Precision | 0.372 | 0.423 | 0.358 | 0.345 |
| Recall | 0.495 | 0.550 | 0.494 | 0.473 |
| F1 | 0.272 | 0.322 | 0.244 | 0.249 |
| K-means ARI | 0.165 | 0.251 | 0.098 | 0.127 |
| K-means AMI | 0.282 | 0.357 | 0.199 | 0.221 |

Table 13: Zero-shot cell type classification and clustering performance of different encoders on the Artery dataset.

| Metric | ada-002 | Ember-V1 | scGPT | all-MiniLM-L12-v2 |
|--------|---------|----------|-------|-------------------|
| Accuracy | 0.916 | 0.924 | 0.949 | 0.890 |
| Precision | 0.874 | 0.885 | 0.920 | 0.809 |
| Recall | 0.819 | 0.823 | 0.894 | 0.798 |
| F1 | 0.839 | 0.847 | 0.904 | 0.799 |
| K-means ARI | 0.358 | 0.490 | 0.533 | 0.346 |
| K-means AMI | 0.564 | 0.671 | 0.704 | 0.524 |

Table 14: Zero-shot cell type classification and clustering performance of different encoders on the Tabula Sapeins dataset.

| Metric | ada-002 | Ember-V1 | scGPT | all-MiniLM-L12-v2 |
|--------|---------|----------|-------|-------------------|
| Accuracy | 0.682 | 0.703 | 0.690 | 0.682 |
| Precision | 0.262 | 0.282 | 0.278 | 0.260 |
| Recall | 0.252 | 0.265 | 0.277 | 0.236 |
| F1 | 0.231 | 0.250 | 0.246 | 0.219 |
| K-means ARI | 0.311 | 0.521 | 0.197 | 0.410 |
| K-means AMI | 0.633 | 0.700 | 0.565 | 0.609 |

## A.2 SCMPT - ADDITIONAL DETAILS AND RESULTS

scMPT is trained using the AdamW optimizer (Chollet et al., 2015) (Loshchilov, 2017) (Kingma, 2014), and an exponential decay learning rate scheduler. The initial learning rate, number of epochs, batch size, and decay rate are selected for each dataset through a grid search. This hyperparameter tuning was conducted for each dataset using a validation split derived from the train set. The dense layers which the encoders feed into each have an output dimension of 4096, and use the ReLU (Agarap, 2018) activation function. The output layer uses a softmax activation function. Results for scMPT were averaged across 5 random seeds, with mean and standard deviation reported as Mean (Standard Deviation).

Table 15: scMPT cell type classification performance vs unimodal models - Aorta dataset.

| Model | Accuracy | Precision | Recall | F1 |
|---|---|---|---|---|
| scMPT | 0.971 (0.0008) | 0.967 (0.0019) | 0.954 (0.0013) | 0.960 (0.0012) |
| scGPT + MLP | 0.968 | 0.960 | 0.949 | 0.954 |
| Ember-V1 + MLP | 0.940 | 0.923 | 0.869 | 0.889 |

Table 16: scMPT cell type classification performance vs unimodal models - Artery dataset.

| Model | Accuracy | Precision | Recall | F1 |
|---|---|---|---|---|
| scMPT | 0.962 (0.00057) | 0.935 (0.00033) | 0.928 (0.0019) | 0.931 (0.0011) |
| scGPT (MLP) | 0.961 | 0.932 | 0.926 | 0.929 |
| Ember-V1 + MLP | 0.949 | 0.910 | 0.899 | 0.903 |

Table 17: scMPT cell type classification performance vs unimodal models - Bones dataset.

| Model | Accuracy | Precision | Recall | F1 |
|---|---|---|---|---|
| scMPT | 0.684 (0.031) | 0.549 (0.0126) | 0.691 (0.0094) | 0.554 (0.0217) |
| Ember-V1 + MLP | 0.674 | 0.526 | 0.686 | 0.541 |
| scGPT (MLP) | 0.630 | 0.509 | 0.657 | 0.498 |

Table 18: scMPT cell type classification performance vs unimodal models - Tsapeins dataset.

| Model | Accuracy | Precision | Recall | F1 |
|---|---|---|---|---|
| scMPT | 0.764 (0.0033) | 0.349 (0.0177) | 0.297 (0.0094) | 0.290 (0.0128) |
| scGPT (MLP) | 0.736 | 0.304 | 0.274 | 0.261 |
| Ember-V1 + MLP | 0.748 | 0.291 | 0.236 | 0.236 |

Table 19: scMPT cell type classification performance vs unimodal models - Myeloid dataset.

| Model | Accuracy | Precision | Recall | F1 |
|---|---|---|---|---|
| **scMPT** | 0.664 (0.0026) | 0.387 (0.009) | 0.364 (0.0086) | 0.3694 (0.0087) |
| **scGPT (fine-tuned)** | 0.642 | 0.366 | 0.347 | 0.346 |
| **scGPT (MLP)** | 0.622 | 0.380 | 0.349 | 0.354 |
| **scGPT (from-scratch)** | 0.606 | 0.304 | 0.339 | 0.309 |
| **Ember-V1 + MLP** | 0.601 | 0.352 | 0.314 | 0.325 |

Table 20: scMPT cell type classification performance vs unimodal models - MS dataset.

| Model | Accuracy | Precision | Recall | F1 |
|---|---|---|---|---|
| **scGPT (MLP)** | 0.845 | 0.769 | 0.735 | 0.726 |
| **scMPT** | 0.837 (0.00089) | 0.733 (0.0039) | 0.718 (0.0024) | 0.704 (0.003) |
| **scGPT (fine-tuned)** | 0.856 | 0.729 | 0.720 | 0.703 |
| **scGPT (from-scratch)** | 0.798 | 0.660 | 0.623 | 0.600 |
| **Ember-V1 + MLP** | 0.687 | 0.597 | 0.582 | 0.568 |

Figures 2 and 3 below summarize scMPT cell type classification performance compared to scGPT; both for the setting where an MLP is trained on top of scGPT, and where k-nearest neighbours is used for zero-shot cell type classification.

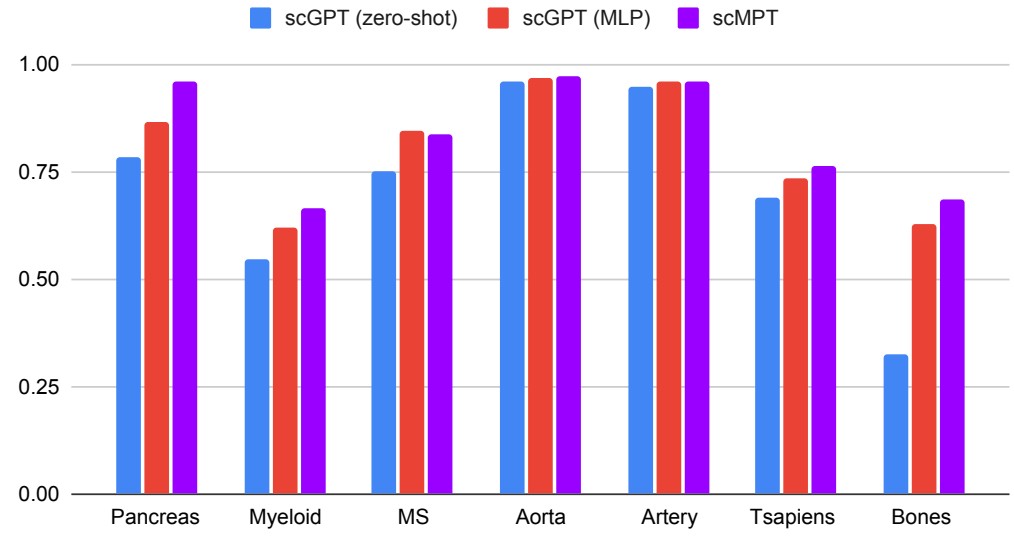

Figure 2: Comparison between cell type classification accuracy of scMPT and scGPT on all datasets studied. scMPT outperforms scGPT on most datasets tested, with particularly significant improvements on the Pancreas and Bones datasets.

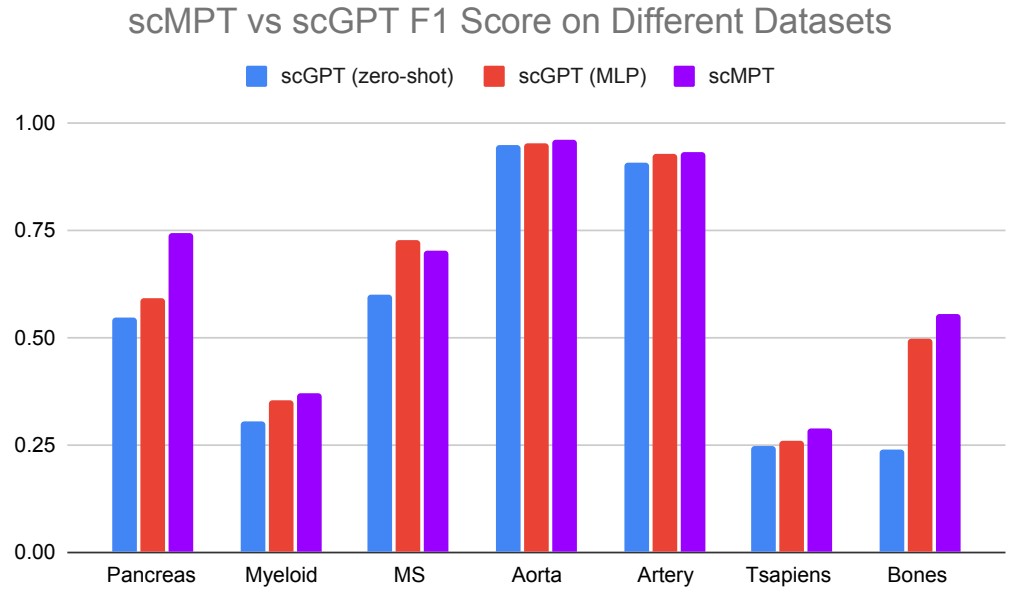

Figure 3: Comparison between cell type classification F1 score of scMPT and scGPT on all datasets studied. scMPT outperforms scGPT on most datasets tested, with particularly significant improvements on the Pancreas and Bones datasets.

### A.3 GPT-4O PROMPTS AND PIPELINE ILLUSTRATION

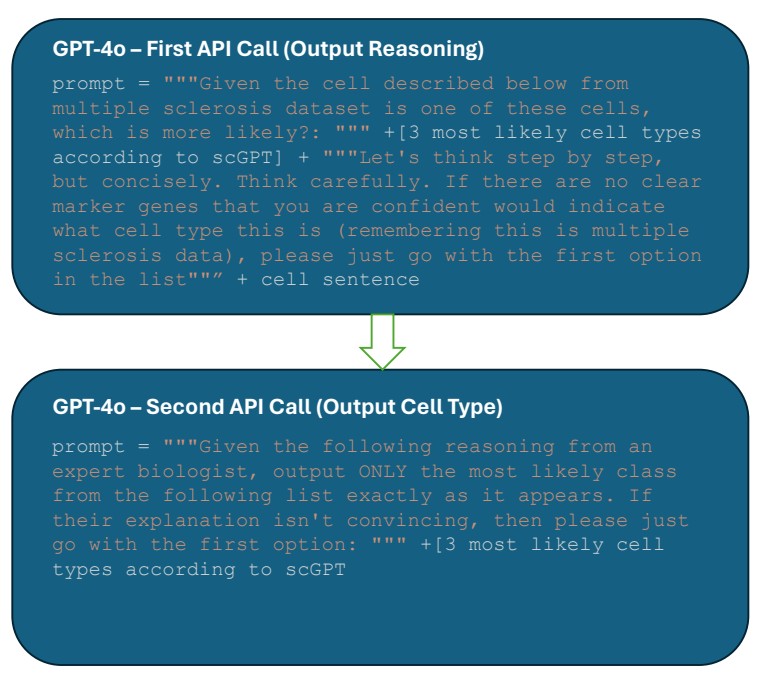

Figure 4: Pipeline for method used to allow GPT-4o to complement scGPT.

## A.4 ADDITIONAL RESULTS FOR ABLATION STUDY

Below are results for all ablations for ada-002 on the Aorta dataset.

Table 21: Ablation Test Results - Aorta Data (zero-shot) - ada-002 encoder

| Metric | ada-002 Gene Ablated | ada-002 Gene+Order Ablation | ada-002 Order Ablation | ada-002 | ada-002 original sentence length |
|---|---|---|---|---|---|
| **Accuracy** | 0.838 | 0.535 | 0.697 | 0.889 | 0.872 |
| **Precision** | 0.883 | 0.253 | 0.483 | 0.916 | 0.865 |
| **Recall** | 0.567 | 0.177 | 0.303 | 0.765 | 0.670 |
| **F1** | 0.615 | 0.171 | 0.311 | 0.804 | 0.716 |
| **k-means ARI** | 0.477 | 0.010 | 0.203 | 0.363 | 0.350 |
| **k-means AMI** | 0.451 | 0.005 | 0.219 | 0.538 | 0.510 |

Next, we present results for the gene name ablation test (where gene names are replaced with SHA-256 hashes) for Ember-V1 on other datasets, reporting classification performance before and after this ablation.

Table 22: Ember-V1 cell type classification performance before and after gene name ablation on datasets used for evaluation in scGPT paper (Cui et al., 2024)

| Metrics | Pancreas Data | | MS Data | | Myeloid Data | |
|---|---|---|---|---|---|---|
| | Ember-V1 Ablated | Ember-V1 | Ember-V1 Ablated | Ember-V1 | Ember-V1 Ablated | Ember-V1 |
| accuracy | 0.9755 | 0.984 | 0.364 | 0.445 | 0.483 | 0.499 |
| precision | 0.803 | 0.796 | 0.341 | 0.456 | 0.317 | 0.333 |
| recall | 0.748 | 0.767 | 0.283 | 0.338 | 0.244 | 0.266 |
| F1 | 0.771 | 0.778 | 0.273 | 0.325 | 0.261 | 0.284 |

Table 23: Ember-V1 cell type classification performance before and after gene name ablation on other datasets.

| Metrics | Tabula Sapeins Data | | Artery Data | | Bones Data | |
|---|---|---|---|---|---|---|
| | Ember-V1 Ablated | Ember-V1 | Ember-V1 Ablated | Ember-V1 | Ember-V1 Ablated | Ember-V1 |
| accuracy | 0.534 | 0.702 | 0.877 | 0.924 | 0.282 | 0.427 |
| precision | 0.217 | 0.288 | 0.805 | 0.885 | 0.424 | 0.421 |
| recall | 0.181 | 0.265 | 0.743 | 0.823 | 0.429 | 0.551 |
| F1 | 0.162 | 0.252 | 0.767 | 0.847 | 0.228 | 0.321 |

