# OpenReview forum: "scMPT: towards applying large language models to complement single-cell foundation models"
_ICLR.cc/2025/Conference — ICLR 2025 Conference Withdrawn Submission_

### Official Review · Reviewer_Fkt2 · 2024-10-30

**Soundness:** 1
**Presentation:** 2
**Contribution:** 2
**Rating:** 1
**Confidence:** 5

**Summary:**

This paper provides a hybrid architecture that integrates the cell representation from pre-trained scGPT and LLM-based text encoders. In detail, the scRNA-seq data is forwarded to scGPT [1] to obtain the cell embedding. Another path converts the data to a so-called cell sentence (this tech was introduced in GenePT [2]) and passes it to an encoder to get the textual feature. The whole framework is simple to understand. Finally, the author conduct experiments to discuss the impact of applying the LLM-based text encoder. Although the performance on cell type annotation seems sound, the other contributions to the biological field are generally few. Besides those, the whole framework just concats the output of scGPT and GenePT, which then passes through a dense layer. The technique contribution is quite weak.

[1] Haotian Cui, et. al. scGPT: toward building a foundation model for single-cell multi-omics using generative AI\
[2] Yiqun Chen, et al. GenePT: A Simple But Effective Foundation Model for Genes and Cells Built From ChatGPT

**Strengths:**

S1. Simple but effective hybrid approach.\
S2. Part of the discussion about biomarkers is interesting.\
S3. The performance on cell type annotation is good.

**Weaknesses:**

W1. The technique contribution is weak. In analogy, the framework is like those earlier NLP approaches, which simply combine the pre-trained language model (such as BERT) for better textual representation. For instance, although ChatNT [1] only employs an existing technique, its novel architecture makes it interesting and insightful for AI researchers or bio researchers. \
W2. This paper lacks a detailed analysis of biological-related tasks. Most of the scXXX aims to simulate some biological signal, e.g., GRN or Aging, but not cell-type annotation. For instance, expression pattern discussion in scGPT [2], cross-species analysis in GeneCompass [3], interpretability, and novel cell type discovered discussion in scBERT [4]. Those ignored deep analyses or insightful discussions limited the contribution of this framework to peer literature. \
W3. Only a few datasets have been selected for this paper. Lack of a general robust analysis among multiple species, tissue types, aging, develop, or a large-scale dataset.\

[1] Guillaume Richard , et al. ChatNT: A Multimodal Conversational Agent for DNA, RNA and Protein Tasks\
[2] Haotian Cui, et. al. scGPT: toward building a foundation model for single-cell multi-omics using generative AI\
[3] Xiaodong Yang, et. al. GeneCompass: deciphering universal gene regulatory mechanisms with a knowledge-informed cross-species foundation model\
[4] Fang Yang, et, al., scBERT as a large-scale pretrained deep language model for cell type annotation of single-cell RNA-seq data

**Questions:**

Please refer to the weakness.

---

### Official Review · Reviewer_WiBH · 2024-10-31

**Soundness:** 2
**Presentation:** 3
**Contribution:** 2
**Rating:** 3
**Confidence:** 4

**Summary:**

The paper introduces a novel model (scMPT) that leverages synergies between single-cell foundation models and large language models (LLMs) to improve the interpretation of single-cell data. The study investigates the integration of biological insights into the performance of LLMs, which is not well understood, with a focus on using LLMs complementarily rather than as alternatives to existing foundation models.

**Strengths:**

The paper introduces scMPT, a model that combines representations from both single-cell models and LLMs to achieve better performance than either model alone across various datasets.

**Weaknesses:**

The paper reveals that LLMs, particularly text encoders, heavily rely on marker genes to predict cell types. This suggests a potential limitation in the model’s understanding of broader biological contexts beyond these markers.

The actual biological factors contributing to the performance improvements seen with LLMs are not clearly defined. While the paper conducts interpretability and ablation studies, it is not definitive about which aspects of biological knowledge are being leveraged.

The paper mainly confirms existing knowledge that marker genes are crucial in cell type classification without introducing new biological insights or innovative ways of interpreting single-cell data using LLMs.

**Questions:**

See weakness.

---

### Official Review · Reviewer_Gxd2 · 2024-11-04

**Soundness:** 3
**Presentation:** 3
**Contribution:** 1
**Rating:** 3
**Confidence:** 4

**Summary:**

The paper studies several interesting problems of single-cell foundation models for cell type annotation. Specifically, the authors present an in-depth analysis of the interpretability of text-based annotation methods, a performance comparison between scGPT and text-based methods, and an ensemble method scMPT for fusing both methods.

The most interesting finding from the paper is that the Ember-V1 effectively leverages the syntactical similarity between cell sentences, instead of solely relying on gene semantics and inherent domain knowledge.

Despite this intriguing finding, the overall contribution and novelty of this paper are very weak. The paper mainly applies existing model interpretability methods to existing cell-type annotation methods and proposes a straightforward ensemble of both methods. Meanwhile, the proposed scMPT's performance mostly lies between GPT4o and scGPT, which is not particularly surprising. The highlighted markers from the interpretability experiment do not provide further biological or practical insights.

In summary, I would consider this paper to be an excellent technical report instead of a scientific paper. I would suggest the authors dig deep into the ablation experiments and propose some alternative methods that specifically address the weak points of existing methods exposed in the preliminary study.

**Strengths:**

1. The paper provides a comprehensive comparison between multiple text encoders for single-cell cell-type annotation.
2. The paper discovers an interesting fact that the text-based methods work through both gene semantic and cell sentence similarity.

**Weaknesses:**

1. The contribution of the methodology in this paper is incremental.
2. The highlighted markers from the interpretability experiment lack the further analysis that can provide biological or practical insights.
3. The proposed scMPT's performance mostly lies between GPT4o and scGPT.

**Questions:**

The authors did not cite top-performing single-cell foundation models other than scGPT.  Is it possible to combine scMPT with other foundation models, e.g., CellPLM[1], scFoundation[2], to achieve higher performance?

[1] CellPLM: pre-training of cell language model beyond single cells, ICLR 2024
[2] Large-scale foundation model on single-cell transcriptomics, Nature Methods 2024

---

### Official Review · Reviewer_w5Ui · 2024-11-05

**Soundness:** 2
**Presentation:** 2
**Contribution:** 2
**Rating:** 5
**Confidence:** 2

**Summary:**

The work presents scMPT, leveraging the strengths of both models, integrating gene expression data from scGPT and biotextual knowledge from LLM to improve the accuracy of cell classification and stability across different datasets. It is shown that scMPT outperforms independent models by capturing complementary insights, particularly in identifying marker genes critical for cell type prediction. This approach opens new avenues for applying LLMs as complementary tools for biological data analysis.

**Strengths:**

- scMPT consistently outperforms standalone models (scGPT and LLMs) by combining complementary insights from text and gene expression data.
- The model effectively identifies biologically relevant marker genes, adding transparency to its cell type predictions.
- The model’s architecture keeps encoders *frozen*, reducing computational costs while still improving accuracy.

**Weaknesses:**

- The study only uses “cell sentences” as the textual representation, potentially missing out on other informative representations.
- In Figure 1 (A depiction of the scMPT architecture), it is shown that all the information comes from the scRNA data. Factors unrelated to biology may contribute to model performance, questioning the biological specificity of the embeddings.

**Questions:**

**Ablation Studies**
- Have you conducted any ablation studies on the information loss from scRNA format to text format?

**Improvement Opportunities**
- I think there is lots of tables exceeding the margin of the page. Pages are P6, P7, P8, P9, P19.

---

### Official Review · Reviewer_vBus · 2024-11-09

**Soundness:** 3
**Presentation:** 2
**Contribution:** 2
**Rating:** 5
**Confidence:** 3

**Summary:**

The manuscript describes several explorations using language models trained on single-cell transcriptomics data. A distinction is drawn between those pre-trained on strictly transcriptomics data ("single-cell foundation models") and those pre-trained on a more general corpus ("large language models"). Insight is sought into the appropriate usage and differential capabilities of these two classes of models.

First, a set of ten pre-trained models are used in a to encode a set of text representations of transcriptomics data ("cell sentences") and the performance of the resulting embeddings evaluated in cell classification and transcriptome clustering tasks. It is found that Ember-V1 largely outperforms other general-purpose pre-trained models, including those used in previous publications, and that the dedicated pre-trained model scGPT outperforms the generalist models as a class.

In a second experiment, the cell classification performance of several models is evaluated with the inclusion of a list of potential cell types included in the prompt. The generalist models show some capability on the task under these experimental conditions, in contrast to previously reported results wherein the potential cell types were not provided.

Third, two methods, integrated gradients and LIME, are used to identify which information in the cell sentences is being utilized during the cell classification task. A set of marker genes are found amongst the top 10 importance-ranked genes.

Next, a set of experiments are run in which the gene names, frequency of occurrence, or both provided to the Ember-V1 model are altered. The resulting embeddings are once again evaluated by performance on cell classification and transcriptome clustering tasks as in the first experiment. Embeddings generated under both obfuscations retain better-than-random performance, suggesting that the cell sentence representation used contains additional information in its syntax/structure.

Finally, a generalist model (Ember-V1) and a specialist model (scGPT) are joined into a single model (scMPT). The larger models' weights are frozen and trainable MLP segments added for joining their embeddings. The joined model is found to perform well on the cell classification task, and the isolated addition of the MLP segment to the output of either Ember-V1 or scGPT is found to improve cell classification performance over utilizing KNN classification with their embeddings.

**Strengths:**

- The manuscript is very aware of the recent literature in the space, covers a breadth of topics, provides sufficient detail to replicate their results, and is generally scientifically sound (save a lack of replicates on most experiments).
- The stated goal of "elucidat[ing] what pre-trained knowledge of biology LLMs leverage when applied to single-cell analysis" is both interesting and potentially useful.

**Weaknesses:**

The primary weakness of the manuscript is a lack of depth on any of the topics it investigates; it feels like a bundling of several fairly surface-level investigations around a loose theme. A couple of the composite activities would make for an interesting study if expanded into a full, depth-first study.
- Understanding how the abstract science information a generalized large language model obtains during pre-training is a fascinating concept, and the finding that information may be encoded in the literal syntax of a
cell sentence is interesting, but it feels like there is more to learn here. Could more be learned about how that information is encoded and utilized with a set of studies targeting explicit manipulation of the syntax?
- The strategy of merging the generalist and specialist models purveys an unstated hypothesis that the two types of models may capture differing information. Could we better understand what each component model is contributing to the performance of the composite model with a set of ablation studies similar to those run in the previous activity?

**Questions:**

- The related work section criticizes the ensembling approach used in the GenePT manuscript, stating that aggregating predictions "ignores possible synergies between different modalities". Was this aggregation strategy benchmarked? It would be helpful to include it in Table 6.
- A hash function is used to obscure the gene names during the ablation study. Why was this strategy chosen as opposed to replacement with a randomized string? Could the hashed names be carrying (permuted) information over into the prompts?

---

### Note · Authors · 2024-11-20

I have read and agree with the venue's withdrawal policy on behalf of myself and my co-authors.